# Spatiotemporal Variation and Driving Forces Analysis of Eco-System Service Values: A Case Study of Sichuan Province, China

**DOI:** 10.3390/ijerph19148595

**Published:** 2022-07-14

**Authors:** Chengjin He, Huaiyong Shao, Wei Xian

**Affiliations:** 1College of Earth Sciences, Chengdu University of Technology, Chengdu 610059, China; 2020020014@stu.cdut.edu.cn; 2College of Resources and Environment, Chengdu University of Information Technology, Chengdu 610225, China; xianwei@cuit.edu.cn

**Keywords:** ecological service values, spatial-temporal variation, tradeoffs and synergies, multi-scale, driving forces

## Abstract

Sichuan Province is an important ecological barrier in the upper reaches of the Yangtze River. Therefore, it is critical to investigate the temporal and spatial changes, as well as the driving factors, of ecosystem service values (ESVs) in Sichuan Province. This paper used land use data from 2000, 2005, 2010, 2015, and 2020 to quantify the spatiotemporal changes in the ESVs in Sichuan Province. Correlation coefficients and bivariate spatial autocorrelation methods were used to analyze the trade-offs and synergies of ESVs in the city (autonomous prefecture) and grid scales. At the same time, we used a Geographical Detector model (GDM) to explore the synergies between nine factors and ESVs. The results revealed that: (1) In Sichuan Province, the ESVs increased by 0.77% from 729.26 × 10^9^ CNY in 2000 to 741.69 × 10^9^ CNY in 2020 (unit: CNY = Chinese Yuan). Furthermore, ecosystem services had a dynamic degree of 0.13%. Among them, the ESVs of forestland were the highest, accounting for about 60.59% of the total value. Among the individual ecosystem services, only food production, environmental purification, and soil conservation decreased in value, while the values of other ecosystem services increased. (2) The ESVs increased with elevation, showing a spatial distribution pattern of first rising and then decreasing. The high-value areas of ESVs per unit area were primarily distributed in the forestland of the transition area between the basin and plateau; The low-value areas were distributed in the northwest, or the urban areas with frequent human activities in the Sichuan Basin. (3) The tradeoffs and synergies between multi-scale ecosystems showed that ecosystem services were synergies-dominated. As the scale of research increased, the tradeoffs between ecosystems gradually transformed into synergies. (4) The main driving factors for the spatial differentiation of ESVs in Sichuan Province were average annual precipitation, average annual temperature, and gross domestic product (GDP); the interaction between normalized difference vegetation index (NDVI) and GDP had the strongest driving effect on ESVs, generally up to 30%. As a result, the distribution of ESVs in Sichuan Province was influenced by both the natural environment and the social economy. The present study not only identified the temporal and spatial variation characteristics and driving factors of ESVs in Sichuan Province, but also provided a reference for the establishment of land use planning and ecological environmental protection mechanisms in this region.

## 1. Introduction

Ecosystem services refer to the products and benefits which humans obtain from ecosystems. The provision of such services can occur directly or indirectly, depending on the structure, processes, and functions of ecosystems [1,2]; ecosystem services are essential for maintaining life on Earth and the ecosystem integrity [3,4]. The Millennium Ecosystem Assessment (MEA) promulgated by the United Nations in 2005 divided ecosystem services into four categories: provisioning services, regulating services, supporting services and cultural services, and quantifying the importance of ecosystems to human well-being is one of its main objectives [5]. Ecosystem service values (ESVs) are a monetary quantification of ecosystem services. In general, scientific evaluation of ESVs is conducive to improve people’s awareness of biodiversity conservation, optimize land use structure, and provide a reference for regional ecological security management and sustainable development [6,7].

There are generally two ways to quantify the ESVs [8]: one is based on the unit price of ecological products, using the shadow engineering method, market price method, carbon tax method, and other methods to calculate the ESVs [9,10,11,12]. This method has high data requirements, complex calculations, and thus a unified and versatile evaluation standard is difficult to achieve. The other is in concert with the economic value of the unit area of the ecosystem, multiplying the value coefficient of the corresponding land use type area to obtain the ESVs [13], this method was proposed by Costanza in 1997, and applied for the assessment of ecosystem services all over the world [3]. However, the method is susceptible to subjective factors and insensitive to the temporal and spatial changes in the properties and quality of ecosystems [14,15]. To realize the dynamic change of ecosystem service values, a dynamic equivalent factor combined with remote sensing was proposed [8]. At present, the ecosystem adjustment coefficient is generally determined by incorporating vegetation coverage [16,17], net primary productivity [18,19,20,21], and normalized vegetation index [22,23], and the calculated ESVs have qualified spatial-temporal resolution and high degree of credibility.

There are various degrees of trade-offs and synergies among ecosystem services due to their complex and dynamic interactions [24]. The enhancement, in term of provision level, of one ecosystem service at the expense of the provision of other ecosystem services is referred to as a trade-off, whereas synergy is the simultaneous increase or decrease of two ecosystem services [25]. With the rapid growth of the global economic population and the growing shortage of resources, the study on ecosystem service trade-offs and synergies not only is of great significance to global environmental changes and improvement of the regional ecological environment but also provides a theoretical basis for the rational development and utilization of resources [26,27]. Therefore, exploring the complex interactions behind ecosystem services has become a popular topic among the scientific communities in the past few years [28]. To date, tradeoffs and synergies among ecosystem services have been analyzed at a global scale [29,30], national scale [31], watershed scale [32,33], and landscape scale [34,35]. However, ecosystem service trade-offs and synergies are dependent on spatiotemporal scales, and the synergistic relationship of ecosystem service tradeoffs at the regional scale is not able to represent the ecosystem service relationship on a small scale [36,37]. The tradeoffs and synergies of ecosystem services vary over time and space. Moreno-Llorca et al. analyzed the relationship between four ecological services in the Sierra Nevada Mountains of Spain from three nested spatial scales of biosphere reserves, watersheds, and grid cell levels [38]. Yang et al. investigated the trade-offs and synergies among ecosystem services in the Yellow River Basin and its eight subbasins. Their findings revealed definite secondary basin differences and regional regularities, implying that tradeoffs and synergies were scale-dependent [39].

Most of the aforementioned studies had carried out detailed research on the evaluation of ESVs and their temporal and spatial changes. Although temporal and spatial changes in ESVs are important, potential factors that affect changes in ESVs still need to be considered. Understanding the ESVs and their driving forces helps to achieve the goal of sustainable development of various ecosystems and the harmonious coexistence of human society and natural ecosystems [40]. At present, some scholars have discussed the relationship between land use change and ecosystem services [1,41,42], and also analyzed the impact of ecological restoration policies on changes in ecosystem services [43,44,45]. However, previous studies have shown that changes in ESVs are the result of a combination of multiple driving factors. Therefore, a comprehensive analysis of the impact of natural factors [46,47,48,49], socio-economic factors [49,50,51], and political factors [44,52] on the ESVs is helpful to understand the ecological environment protection and formation mechanism.

Sichuan Province is located on the eastern edge of the Qinghai-Tibet Plateau. The region has large undulating terrain, complex geological structure, frequent natural disasters, and sensitive and fragile ecosystems. Affected by human factors, the ecological environment has been seriously degraded. In addition, it has been successively included in the “Returning Farmland to Forest Project”, “Returning Grazing to Grassland”, the Qinghai-Tibet Plateau Region (Sichuan Province) Ecological Construction and Environmental Protection Planning, and other ecological projects and planning to protect and construct the ecological environment. Therefore, it is very important to fully understand the temporal and spatial variation characteristics of ESVs in Sichuan Province and their influencing factors for regional ecological environment protection and sustainable development.

Previous studies have analyzed the relationship between land use change and ESVs on the one hand [53,54,55], and the relationship between regional ESVs and driving factors on the other hand [56,57]. Therefore, the value coefficient and evaluation model were modified according to the actual situation in the study area. Based on the land use data in 2000, 2005, 2010, 2015, and 2020, the temporal and spatial variation characteristics and influencing factors of ESVs were analyzed. The main objectives of this study were: (1) to quantify ESVs and reveal the spatial distribution characteristics of ESVs; (2) to identify trade-offs and synergies between the values of individual ecosystem services through correlation analysis; (3) to use bivariate spatial autocorrelation analysis at different scales to reveal the spatial heterogeneity of trade-offs and synergies among the six groups of ecosystem services; (4) to quantify the degree of impact of driving factors on ecosystem service value.

### Study Area

Our study area is located at the intersection of the Qinghai-Tibet Plateau and the Middle-Lower Yangtze plain. Sichuan Province (26°03′~34°19′ N, 97°21′~108°12′ E) covers an area of484,000 km^2^. The landform of Sichuan Province varies greatly from east to west, the terrain is complex and diverse, and the terrain is high in the west and low in the east (Figure 1). The western part is plateau and mountainous, and the altitude is mostly above 3 km. The eastern part is a basin and a hill, and the altitude is mostly between 0.5 and 2 km. Sichuan Province has three major climates: The subtropical humid and semi-humid climate in the Sichuan Basin is, respectively, divided into four distinct seasons, with the same period of rain and heat, the average annual temperature is 16~18 °C, with 1000–1200 mm of precipitation. The subtropical semi-humid climate in the mountains of southwest Sichuan is not clearly distinguished between the four seasons, the annual average temperature is 12~20 °C, with 900–1200 mm of precipitation. The alpine plateau in northwest Sichuan has an alpine climate with a great difference in altitude and significant temperature changes. The average annual temperature is 4~12 °C, with 500–900 mm of precipitation.

## 2. Materials and Methods

### 2.1. Factor Selection and Setting

We referred to the research of Xie et al. on the equivalent scale of ecological services per unit area of terrestrial ecosystems in China [58] and combine the natural and socioeconomic conditions of the study area to classify ecosystem services into four categories: provisioning, regulating, supporting and cultural services, and further subdivided into 9 services (Table 1). According to previous studies, it was determined that the farmland equivalent factor of Sichuan Province is 1.35 times that of the national farmland [59]. The average value of the coniferous forest and shrub forest was selected for forestland. The equivalent factors of grassland and wetland were set according to the research results of Zheng [60]. The setting of the equivalent factor of bare land and construction land is based on the research results of Li [18].

### 2.2. Data Sources and Processing

The land use data of five periods (2000, 2005, 2010, 2015, and 2020) were obtained from the 30 m resolution annual China land cover dataset (CLCD) [61], Then it was divided into 7 categories: farmland, forestland, grassland, water, bare land, construction land, and wetland. Based on the MODIS dataset in the Google Earth Engine platform. The Modified Normalized Difference Water Index (MNDWI) and the Remote Sensing Ecological Index (RSEI) were calculated in the study area. Regardless of price fluctuations, according to the website of the State Bureau of Grain and Material Reserves (http://www.lswz.gov.cn (accessed on 12 January 2022)) and the Sichuan Statistical Yearbook, the average price of grain in 2010 was 1.87 (unit: CNY/kg). The study area was divided into 1 km grids, and the land use data of each grid and 21 prefecture-level administrative districts were extracted based on the land use data of five periods.

Considering the availability of data and the fact that changes in ESVs were influenced by a variety of factors such as the natural environment and the social economy, this paper identified the following nine driving factors: Elevation and slope were derived from Geospatial Data Cloud (http://www.gscloud.cn (accessed on 20 January 2022)). The annual average temperature, annual average precipitation, and soil organic carbon content were obtained from the National Tibetan Plateau Data Center (http://data.tpdc.an.cn (accessed on 15 February 2022)). Population density, gross domestic product (GDP), normalized difference vegetation index (NDVI), and river data were from the Resource and Environmental Science and Data Center (https://www.resdc.cn/ (accessed on 10 March 2022)). Finally, we analyzed the spatial differentiation characteristics between them and ESVs.

### 2.3. Ecosystem Service Values

#### 2.3.1. Ecosystem Service Assessment Model

Ecological service values are dynamic value that changes over time and varies with the type, size, and quality of regional ecosystems. Considering the impact of temporal and spatial changes in ecological quality on ESVs, RSEI and MNDWI were chosen to correct the ESVs of each pixel at each moment. The formula is as follows:(1)$$ESV\left(Si,j,t_{h}\right)=\overset{q}{\underset{f=1}{\sum }}VC_{if}\times A_{Si}\times R\left(Si,j,t_{h}\right)$$

In Formula (1), when the i-th pixel is the j-th land use type, $ESV\left(Si,j,t_{h}\right)$ represents the ESV at the research moment $t_{h}$. The value coefficient of the f-th ecosystem service function of the j-th land use type is denoted by $VC_{if}$ (CNY/ha). $A_{Si}$ represents the pixel area (ha). $R\left(Si,j,t_{h}\right)$ is the ecological quality correction coefficient at the research moment $t_{h}$ when the i-th pixel is of the j-th land use type.
(2)$$VC_{if}=E_{if}\times C_{crop}\;$$
where $E_{if}$ is the equivalent coefficient of the f-th ecosystem service function of the j-th land use type, representing the weight coefficient of each ecosystem service value. The standard equivalence coefficient $C_{crop}$ (CNY/ha) is based on the ecological service equivalence table per unit area of China’s ecosystems, combined with the social and economic development, the economic value of the natural ecosystem is 1/7 of the food production service value provided by the existing unit area of cultivated land without human input. The economic value of an equivalent factor of ecological service value in Sichuan Province is calculated to be 1403.56 CNY/ha.
(3)$$R\left(Si,j,t_{h}\right)=\frac{e\left(Si,j,t_{h}\right)}{\sum_{i=1}^{n}e\left(Si,j,t_{h}\right)/n}$$
where $e\left(Si,j,t_{h}\right)$ is the ecological condition index of the j-th land use type of the i-th pixel when the study year is $t_{h}$. $\overset{n}{\underset{i=1}{\sum }}e\left(Si,j,t_{h}\right)/n$ is the average value of the ecological condition index of all pixels of the same land type at the same time.

RSEI is to reflect the impact of changes in external factors such as human activities, climate change, and environmental state changes on the environment. In addition to quantitatively evaluating the ecological quality of the area, RSEI can also visualize the ecological environment of the study area, and support the analysis, prediction, and assessment of temporal and spatial changes in the ecological environment quality of the study area [62]. MNDWI can quickly extract water body information [63]. Li et al. corrected the equivalent factor pixel by pixel through RSEI and MNDWI, which can effectively display the temporal and spatial changes of ESVs in each pixel [64]. In order to better distinguish the ecological status between pixels, this paper introduced RSEI and MNDWI to construct $e\left(Si,j,t_{h}\right)$.
(4)$$e\left(Si,j,t_{h}\right)=RSEI+MNDWI$$

The formula for calculating MNDWI is as follows (5):(5)$$MNDWI=\frac{\rho_{green}-\rho_{mir}}{\rho_{green}+\rho_{mir}}$$

RESI is defined as a function of greenness, wetness, heat, and dryness components, where greenness uses the normalized vegetation index (NDVI) to describe the growth and change of regional vegetation; The land surface temperature (LST) obtained by thermal infrared remote sensing inversion represents heat; The land surface moisture (LSM) is represented by the wetness component obtained by the tasseled-cap transformation of the multispectral image; Normalized differential build-up and bare soil index (NDBSI) composed of the index-based built-up index (IBI) and bare soil index (SI) was selected to indicate dryness.
(6)$$RSEI=f\left(Greeness,Wetness,Heat,Dryness\right)$$

Then, the four indicators such as NDVI, LSM, LST, and NDBSI are normalized to be between 0 and 1; Secondly, perform principal component analysis on the multi-band images synthesized by the four indicators, using the first principal component (PC1) as the starting remote sensing ecological index $RSEI_{0}$; Finally, the RSEI obtained by normalizing $RSEI_{0}$ ranges from 0 to 1. The larger the RSEI value, the better the ecological condition [65,66].

#### 2.3.2. Dynamic Degree of Ecological Service Values

The dynamic changes in regional ecological service values were analyzed using the dynamic degree of ESVs, following the formula below:(7)$$K_{esv}=\frac{ESV_{b}-ESV_{a}}{ESV_{a}}\times \frac{1}{T}\times 100$$
where $K_{esv}$ reflects the intensity of ecosystem service values changes with time, *T* represents the period, $ESV_{a}$ is the initial ecological service value in a period, and $ESV_{b}$ is the value of terminated ecological service within a period.

### 2.4. Methods of Analysis

#### 2.4.1. Correlation Analysis

Correlation analyses were conducted to determine whether there were synergies or trade-offs between these ecosystem services, according to Formula (8). The higher the value, the stronger the correlation between the two.
(8)$$V_{xy}=\frac{\sum_{i=1}^{n}\left(x_{i}-\bar{x}\right)\left(y_{i}-\bar{y}\right)}{\sqrt{\sum_{i=1}^{n}{\left(x_{i}-\bar{x}\right)}^{2}}\sqrt{\sum_{i=1}^{n}{\left(y-\bar{y}\right)}^{2}}}$$
where $V_{xy}$ represents the correlation coefficient between the two ecosystems. n represent the total number of ecosystem services. The value of ecosystem services is represented by $x_{i}$ and $y_{i}$, with $\bar{x}$ and $\bar{y}$ being the averages of the corresponding ecosystem service value.

#### 2.4.2. Bivariate Spatial Autocorrelation Analysis

Spatial autocorrelation includes global autocorrelation and local autocorrelation and is mainly used to describe whether the spatial distribution between variables is clustered. In order to describe the correlation between multiple variables, Anselin et al. proposed a bivariate spatial autocorrelation based on the Moran index to reveal the correlation characteristics of the spatial distribution of different elements [67]. This method was introduced into ecosystem services, and GeoDa software was used to calculate the Moran index to evaluate the correlation between ecosystem services, and the local indicators of spatial association (LISA) were used to measure whether the ecosystem services have agglomeration.

#### 2.4.3. Geographic Detector Model

The geographic detector model includes four types of detectors that are used as a new statistical method to investigate the spatial and temporal differentiation characteristics of things and their driving factors. Its central idea is based on the assumption that if an independent variable has a significant influence on a dependent variable, the independent variable’s spatial distribution and the dependent variable’s spatial distribution should be consistent [68]. The dominant factors and their interactions in the spatial differentiation of ESVs in Sichuan Province were analyzed using factor detection and interaction detection in this paper. The following is the formula:(9)$$q=1-\frac{1}{N\sigma^{2}}\overset{L}{\underset{h=1}{\sum }}N_{h}\sigma_{h}^{2}$$

In the Formula (9), q represents the explanatory power of the influencing factors on the spatial differentiation characteristics of ecosystem service value, and its value range is [0, 1]. The greater the value, the greater the interpretive ability of the independent variable X to the dependent variable Y. On the contrary, it is smaller; L is the number of categories of variable Y or driving factor X; N and $\sigma^{2}$ represent the total number of samples in the study area and the discrete variance of the entire area, respectively; $N_{h}$ and $\sigma_{h}^{2}$ represent the number of samples and the dispersion variance in the h area.

## 3. Results

### 3.1. Spatial-Temporal Changes in Landuse

The main land use types in the study area were forest land, followed by grassland and cropland (Figure 2). From 2000 to 2020, among all land use types, forest land increased significantly by 2.06% (Table 2). In the past 20 years, due to urban development, the situation of cropland occupation was more significant, the cropland area had decreased by 8377 km^2^, but the area of impervious land had increased by 1926 km^2^. With the development of animal husbandry, overgrazing led to a significant reduction in the grassland area in the study area, by 5763 km^2^.

### 3.2. Spatial and Temporal Changes in Ecological Services

#### 3.2.1. Characteristics of Temporal Development

According to Formulas (1)–(6), the ESVs of seven land use types in Sichuan Province from 2000 to 2020 were calculated. The calculation results of grid data in the study area were counted and shown in Table 3 and Table 4. The ESVs and dynamic degree of all cities (autonomous prefectures) were calculated by Formulas (1)–(7), and the results were shown in Figure 3.

According to Table 4, total ESVs increased from 724.13 × 10^9^ CNY in 2000 to 729.71 × 10^9^ CNY in 2020, with an average annual growth rate of 0.15%, a dynamic degree of 0.13%, and an increased rate of 0.77%. Among them, the most prominent ecological service was the regulating service, accounting for 65.19% of the total ESVs, followed by the supporting service, the provisioning service, and the cultural service.

In the provisioning services, the ESVs of food production and raw material production were low, showing four stages: gradual decline, rapid decline, slow rise, and rapid rise. In the regulating services, the average annual growth rate (0.46%) and the growth rate (2.3%) of climate regulation were the largest, with an increase of nearly 4.0 × 10^9^ CNY in 20 years, and the minimum growth rate from 2010 to 2015 was only 0.06%. The ESVs of hydrological adjusting and environmental purification showed an upward trend, with a decline rate of 0.13 and 0.14% from 2010 to 2015, so the ESVs reached their peak in 2010. In the supporting services, soil conservation and biodiversity accounted for a similar proportion, but the two trends were opposite, with increases of −0.16 and 1.46%. In terms of cultural services, the ESVs of aesthetic landscapes were relatively low, showing a steady upward trend with an increase of 3.42%. In summary, the four ecosystem services in the study area are mainly regulating services and supporting services. In contrast, provisioning services and cultural services accounted for a smaller proportion but were more variable during the study period. The ESVs of climate regulation, hydrological adjusting, and soil conservation were higher in terms of individual ecological services.

Over the past 20 years, only ESVs in food production, environmental purification, and soil conservation have declined, while other services have increased.

The results in Table 4 showed that among the seven land types in the study area, forest land had the highest ESVs, followed by farmland, grassland, water, wetland, and bare land, and the smallest was construction land. From 2000 to 2020, except for farmland, grassland, and construction land, the ESVs of all other land types increased, but the degree of ESVs change was different for each type of land. Forestland, farmland, and grassland were the main contributors to ESVs in Sichuan Province, and the contribution rate of forest land remains above 58%. The range of ESVs changes in water and wetland was more obvious. The ESVs of water reduced by 14.65% between 2010 and 2020. The ESVs provided by water reached a high of 31.83 × 10^9^ CNY in 2010. Wetland ESVs increased, and the growth rate reached the maximum (1.51%) from 2015 to 2020 and increased to 0.26 × 10^9^ CNY. The proportion of grassland ESVs has decreased by 0.8% over the last 20 years. Bare land accounted for a small proportion of 0.03%, and ESVs fluctuations were small.

Figure 3 reflected the changes in ESVs in cities (autonomous prefectures). Among them, three autonomous prefectures (Ganzi Tibetan Autonomous Prefecture, Liangshan Yi Autonomous Prefecture, and Aba Tibetan Autonomous Prefecture) had relatively high ESVs. In the past 20 years, the ESVs of each region have increased or decreased to varying degrees. Overall, the most dynamic cities were Panzhihua, Suining, Guangyuan, and Luzhou, while Meishan was the least dynamic (−0.04%). From 2000 to 2005, the ESVs of Ziyang, Liangshan, Mianyang, Chengdu, Aba, Ya’an, and Panzhihua decreased by 0.02, 0.06, 0.22, 0.28, 0.38, 0.46 and 3.54%, other regions had increased, and the largest growth rate was Luzhou (3.41%). From 2005 to 2010, the ESVs of 14 cities (autonomous prefectures) in Sichuan Province declined, among which Chengdu had the largest decline (1.82%), and among the remaining areas, the ESVs of the Ganzi had the largest increase of 1.69%; From 2010 to 2015, except for Neijiang, Zigong, Liangshan, Panzhihua and Ganzi, the overall ecology has improved. From 2015 to 2020, the ESVs of Aba and Ganzi increased by 1.47 and 1.14%, respectively, and the ESVs of the rest of the regions showed a downward trend. Overall, the cities (autonomous prefectures) with the largest ESVs fluctuations during the study period were Garze, Luzhou, Panzhihua, and Guangyuan.

#### 3.2.2. Spatial Distribution Characteristics

According to Formulas (1)–(6), the ESVs of the 1-km grid in Sichuan Province from 2000 to 2020 were calculated. GIS software was used for spatial mapping, and the ESVs were divided into 8 levels by the natural breakpoint method. Finally, the ESVs of each city (autonomous prefecture) in Sichuan Province during the study period were obtained. The spatial distribution of ESVs was shown in Figure 4.

In 2000, the high-value areas of the unit area ESVs were mainly distributed in the transition zone between the Sichuan Basin and the Western Sichuan Plateau, and the Zoige Plateau in the northern part of Aba Tibetan and Qiang Autonomous Prefecture. The low-value areas were mainly distributed in the northwest area and the central urban area of Chengdu. The unit area ESVs of the eastern cities were generally high, ranging from 12,500–18,000 CNY/ha. In 2005, the unit area ESVs of Aba Tibetan and Qiang Autonomous Prefecture increased, but there was no significant change in other areas. The ESVs per unit area increased in all cities in 2010, particularly in eastern Sichuan Province. The ESVs per unit area decreased between 2015 and 2020. Compared with the initial stage of the study, the unit area ESVs in the central region showed a downward trend, from 11,400 CNY/ha to 12,500 CNY/ha. Among them, the lowest value of Chengdu and its surrounding cities exceeded 6250 CNY/ha, and the area was constantly expanding. Overall, most of the high-value areas were located in forestland, and gradually decreased to both sides with the change of altitude. The low-value areas were mostly grassland and bare land in the northwest. Compared with the initial period, the unit area ESVs showed an upward trend, and the high-value area expanded at the end of the study.

### 3.3. Tradeoffs and Synergies Analysis

#### 3.3.1. Correlation Analysis of Ecosystem Services

We analyzed the correlation between individual ecosystem services by using Formula (8), and the results were shown in Table 5.

Overall, various ecosystem services were positively correlated at the 0.01 significance level, accounting for 83%, and synergy was the dominant relationship of ecosystem services in Sichuan Province. In the provisioning services, food production was negatively correlated with raw material production, gas regulation, climate regulation, hydrological adjusting, biodiversity, and aesthetic landscape, while positively correlated with other ecosystem services. Among them, the positive correlation between environmental purification and raw material production was weak. In the regulation services, except for the weak correlation among gas regulation, climate regulation, and environmental purification, other ecosystem services showed a significant positive correlation. In the supporting services, there was a weak negative correlation between biodiversity and food production, and environmental purification and biodiversity were positively correlated with the other ecological services, of which biodiversity was closely related to various other services. In terms of cultural services, a positive correlation was found between aesthetic landscapes and various ecosystem services, except food production.

#### 3.3.2. Analysis of Multi-Scale Tradeoffs and Synergies in Ecosystem Services

To further understand the relationship between ecosystem services in Sichuan Province, we analyzed the bivariate spatial autocorrelation between six pairs of ecosystem services at the city (autonomous prefecture) and 5 km grid scales based on GeoDa software. When Moran’s > 0, it means a positive correlation, which is a synergistic effect; when Moran’s < 0, it means a negative correlation, which is a tradeoff effect.

The results in Table 6 showed that the global autocorrelation indices between the six pairs of ecosystem services on the two scales were positive and passed the 5% significance level test. The results showed that there are synergistic effects among the four ecosystem services. In Figure 5 and Figure 6, the characteristics of high-high aggregation and low-low aggregation indicated that the two ecosystem services exhibited synergistic effects, and high-low aggregation and low-high aggregation represented the trade-off effect between the two. At the city (autonomous region) scale, there were trade-off effects between provisioning services and cultural services, regulating services and cultural services, provisioning services, and supporting services in Luzhou. However, there was a trade-off effect for a large number of regions at the grid-scale. Based on the two scales, the correlation between ecosystem services was concluded as follows: as the research scale became larger, the scope of synergistic effects gradually expanded, and the trade-off effect was gradually transformed into a synergistic effect.

At the city (autonomous prefecture) scale, six pairs of ecosystem service functions were distributed in the western and central parts of the study area. However, there were tradeoffs between provisioning and supporting services, provisioning and cultural services, and regulating and cultural services, respectively, in Luzhou. The specific performance was as follows: In addition to regulating and supporting services, the ecosystem services of Ganzi Tibetan Autonomous Prefecture and Liangshan Yi Autonomous Prefecture showed low-low clustering. High-high clusters were distributed in different cities but mainly concentrated in the central part of Sichuan Province. For example, the high-high clusters of provisioning and regulating services, and provisioning and cultural services were mainly distributed in Meishan, Neijiang, Ziyang, and Suining (Figure 5A,C); the high-high clusters of regulating and cultural services, and supporting and culture were mainly distributed in Meishan, Neijiang, and Ziyang (Figure 5E,F); the high-high clusters of provisioning and supporting services were distributed in Nejiang, Ziyang and Suining (Figure 5B); while the high-high clusters of regulating and supporting services were mainly distributed in the western Ganzi Tibetan Autonomous Prefecture and Liangshan Yi Autonomous Prefecture in the western part of the study area (Figure 5D).

At the grid-scale, the spatial distribution characteristics of the tradeoffs and synergies among the six pairs of ecosystem services were significantly different. Synergies were dominant, and tradeoffs existed in a few grids. The results in Figure 6 showed that the synergies among ecosystem services were distributed in blocks, while the distribution of tradeoffs was more scattered. The low-high clusters were mainly distributed around the high-high clusters, and the high-low clusters were distributed in a ring around the low-low clusters. Compared with the city (autonomous prefecture) scale, the synergistic effect at the grid-scale was distributed in the northwest of the study area, or the transition zone between the Sichuan Basin and the Western Sichuan Plateau. The results were as follows: the synergies of provisioning and regulating services, provisioning, and supporting services, and regulating and supporting services were distributed in the northwest of the study area, the area around the Sichuan Basin, and a small part of Chengdu (Figure 6A,B,D). The high-high clustering areas of the six pairs of ecosystem services were mainly distributed in forestland. The tradeoff effects of provisioning and regulating services, provisioning, and cultural services, and regulating and cultural services were more significant. Among them, high-low clusters were distributed around low-low clusters, mainly concentrated in low-altitude areas (Figure 6A,C,F).

### 3.4. Driving Force Analysis

#### 3.4.1. Single Factor Detection of Ecosystem Service Value

The geographic detector model was implemented based on the GD package in R. To make the calculation simple and combined with the actual situation of the study area, we selected nine influencing factors related to the natural environment and social economy to study their driving force on ESV. Among them, X1, X2, X3, X4, X5, X6, X7, X8, and X9 represent elevation, slope, annual average precipitation, annual average temperature, NDVI, and distance from the river, population density, GDP, and soil organic carbon content, respectively. The geographic detector required the independent variable X to be a discretized variable, so it is necessary to discretize the driving factor data. The GD package provides 6 discretization methods, and the optimal method and optimal classification of the data discretization are determined by algorithms, so as to obtain the most explanatory q-value. The results of the factor detector detection are shown in Table 7.

Among them, factors such as temperature, elevation, and GDP had a relatively large contribution rate and were the main driving factors; while precipitation, population density, soil organic carbon, etc. had relatively small contribution rates to the spatial differentiation of ESVs and were secondary driving factors. From the perspective of the impact of driving factors on ESVs, temperature changes within a certain range promote the growth of vegetation, and ESVs increased accordingly. The spatial differentiation of ESVs was influenced by elevation. Lower elevations were more conducive to agricultural development and urban expansion, and ecological land was significantly destroyed, whereas higher elevations had fewer human activities, but the ecological environment for vegetation growth was harsh, and ESVs were also relatively low; GDP can reflect the strength of human activities, and areas with higher GDP value had lower ESVs and vice versa.

There are some differences in the explanatory power of each factor for ESVs across years, but it is generally consistent. Due to the proposal of the western development strategy at the end of the 20th century, the industrial scale of Sichuan Province continued to expand from 2000 to 2010, and the level of environmental pollution was relatively high, so the explanatory power of most driving factors to ESVs was weakened. After the “Eleventh Five-Year Plan”, Sichuan Province has stepped up efforts to protect the ecological environment, and the contradiction between man and nature has been gradually eased. Therefore, after 2010, the ability of each driving factor to explain ESVs in the study area has steadily increased.

#### 3.4.2. Interaction Factor Detection of Ecosystem Service Values

The interaction detector is used to test the interaction between the two influencing factors, that is, whether the two factors will increase or decrease the explanatory power of the ESVs when they act together. The interaction of nine influencing factors on ESVs was obtained with the help of the interaction detector module. The interactive detection results were shown in Figure 7. All driving forces interacted to improve the spatial distribution and differentiation of ESV, and the effects were not independent. This demonstrated that the interaction of multiple factors affected ecosystem services in Sichuan Province from 2000 to 2020. The strongest interaction was between NDVI and GDP, which typically reached 30%. The interaction between the slope and the distance from the river was the weakest, accounting for less than 4% of the total; it can be found that the interaction of elevation, GDP, and other factors had a greater impact on the distribution of ESVs in Sichuan Province. The distribution of land use was influenced by elevation. The intensity of human activities was closely related to the level of GDP value. As a result, the interaction between the natural environment and the social economy influenced the distribution of ESVs in Sichuan Province.

## 4. Discussion

### 4.1. Spatial-Temporal Variation of Ecosystem Services

This paper quantified the ESVs in Sichuan Province in 2000, 2005, 2010, 2015, and 2020 based on land use data. The rational planning and utilization of land resources were of great significance to the ecological environment protection and sustainable development of Sichuan Province. In this paper, the RSEI and MNDWI indices were introduced to correct the ESVs of the study area, and the results showed that the total ecosystem service values in Sichuan Province had improved. The calculation results of ESVs deviated from previous studies because this paper took into account differences in ESVs between different pixels of the same land type, but the changing trend was essentially the same [53]. The results showed that the spatial heterogeneity of ecosystem services is closely related to the spatial distribution of land use [69,70,71]. In recent 20 years, the area of forestland and impervious land in Sichuan province had increased, while the area of cropland had decreased, which was in line with the trend of increasing forest resources and expanding building area in China, and was related to the project of “returning farmland to the forest”. Forestland and grassland cover about 70% of the total area of Sichuan province, leading to the change of regional ESVs. However, the stable distribution of land use types did not significantly change the spatial pattern of ecosystem services.

In response to the current situation of the reduction of grassland and cultivated land and the increase of forest land and construction land in Sichuan Province, the western plateau region should vigorously implement grassland protection systems such as grazing prohibition and fallow, the balance between grass and livestock, and grassland ecological compensation [72]. The eastern region should take advantage of its unique location in the core of the Chengdu-Chongqing economic circle, and actively integrate into the construction of the “Belt and Road” and the Yangtze River Economic Belt, focusing on environmental improvement, and in the process of improvement Repair the environment, protect the ecology during development, comprehensively improve the ecological environment, and achieve green development [73].

### 4.2. Scale Effects of Tradeoffs and Synergies

The tradeoffs and synergies of ecosystem services are spatially heterogeneous and temporally dynamic and change over time and space. The correlation coefficient and Moran index can reveal the trade-offs and synergies of ecosystem services on the temporal and spatial scales. The correlation coefficients of the nine ecosystem services quantified tradeoffs and synergies over time, with fast feedback on raw material production and biodiversity services. However, the cycle of environmental purification was long, and there was a lag in other ecosystem services. The results showed that the correlation coefficient between gas regulation, climate regulation, and raw material production was close to 1, which indicated that vegetation in the study area had a regulating effect on gas and climate, and could also promote the production of raw materials. They had a strong synergistic effect on mutual promotion. On the contrary, the tradeoff between food production and other services was weak, indicating a conflict between food production and environmental protection, reflecting the competition between cultivated land and other land uses [74].

A bivariate spatial autocorrelation method was used at the city (autonomous prefecture) and grid scales to quantify the spatial synergy and tradeoff effects among six pairs of ecosystem services. The results showed that the Moran’s I of the six pairs of ecosystems on both scales were all positive, indicating that the relationship between ecosystem services in Sichuan Province was mainly determined by synergistic effects (Table 6). At the same time, three conclusions can be drawn from the results of the binary space autocorrelation analysis: Firstly, the Moran’s I obtained at the grid-scale was generally larger than the Moran’s I calculated at the city (autonomous region) scale. Secondly, the tradeoff effects among the six pairs of ecosystems at the grid-scale were distributed around the synergistic effect. However, there was not necessarily a trade-off effect between ecosystem services at the city scale, indicating that the tradeoffs and synergies of the same ecosystem services at different regions and scales were also different [19]. Finally, the tradeoffs and synergies among ecosystems at the city scale were mainly distributed in Garze Tibetan Autonomous Prefecture, Liangshan Yi Autonomous Prefecture, and some eastern regions. On the grid-scale, tradeoffs and synergies were only distributed in the northwestern part of the study area, as well as in the transition area between the basin and the western Sichuan Plateau (Figure 5 and Figure 6).

### 4.3. Driving Factor Analysis

Exploring the relationship between ESVs and driving factors provides a basis for ecosystem service management and decision-making. This study quantitatively analyzed the relationship between ESVs and driving factors in the study area and identified the interaction between factors. Ecosystem services in Sichuan Province were the result of the interaction between natural and human factors. However, due to the large proportion of forest land and grassland in the study area, our research focused on the influence of natural factors. The results showed that the annual average temperature was the main driving force, which was consistent with the existing research on the driving force of ecosystem services [48,75,76,77]. The second most important driving factor was the elevation. The spatial heterogeneity of elevation leads to changes in the regional ecological environment, thereby changing the type of land use, which in turn affected the value of regional ecosystem services [46,78].

The spatial differentiation of ESVs in the study area was caused by the interaction of multiple factors. Only analyzing the impact of a single driving factor on ESVs was not able to reveal the contribution of the synergistic effect of driving factors on ESVs. The results of this study showed that the contribution of the interaction of GDP and other drivers to ESVs was generally higher than that of GDP alone to ESVs. Therefore, it is crucial to understand the impact of the interaction between drivers on ESVs. For example, Pan et al. found the synergy of human activities, landscape pattern changes, and natural factors led to the spatial differentiation of ESVs in the study area [79]; Fang et al. explored the impact of natural and anthropogenic factors on the ecosystem service values of the Yangtze and Yellow River basins using a geographically weighted regression model and a geographic detector model, and the results showed that the combined effect of driving factors was much higher than the individual effect [80].

### 4.4. Uncertainties and Further Work Outlook

The interaction mechanisms of ecosystem services are complex, and their evaluation relies on the assessment of similar biological communities, but large ecosystems contain diverse communities and habitat types. Therefore, it is impossible to accurately quantify ESVs. In this study, the secondary land use types of forest land in Sichuan Province were combined, and the equivalent factor was set as the average value of coniferous forest and shrub forest, and the subcategories of land use types were not evaluated in detail.

Although the equivalence factors were revised according to the actual situation of the study area, RSEI was sensitive to phenological changes. Therefore, different image acquisition times will affect the calculation accuracy of RSEI. Considering the high spatial correlation of RSEI, Zhu et al. calculated the remote sensing ecological index based on a moving window, reducing the impact of long-distance features on specific research blocks [81]. Furthermore, this study analyzed the impact of driving factors on the ESVs; however, because the selection of driving factors focused on natural factors, the analysis’ results were biased. Human disturbance factors [82,83,84], natural factors [47,85], socioeconomic factors [51,57], and policy factors [52] also affect the correlation between ecosystem services. Therefore, the spatial aggregation distribution of ecosystem service trade-offs and synergies will be different. Quantifying and modeling tradeoffs and synergies at multiple scales is an important part of ecosystem services research. Therefore, it is necessary to model the tradeoffs and synergies between climate change, land use change, human activity impacts, policy changes, and ecosystem services in future research, as well as forecast future spatiotemporal changes in regional ecosystem services.

## 5. Conclusions

Based on the evaluation model of ecosystem services from 2000 to 2020, nine kinds of ESVs in cities (autonomous prefectures) and 1 km grids were obtained in Sichuan Province. The results revealed that the total amount of ecosystem services increased by 5.58 × 10^9^ CNY, indicating that ecosystem services have improved. The spatial heterogeneity of ESVs was significant, and the ESVs showed a spatial pattern of first increase and then decrease with the increase of altitude. However, due to the stability of the ecosystem structure, the ecosystem pattern in Sichuan Province has not changed significantly in the past 20 years. The results of correlation analysis showed that the synergistic effect of ecosystem services dominated, and only food production and other services showed a weaker tradeoff effect. Bivariate spatial autocorrelation analysis showed that the four main services had different degrees of synergy at different scales, and the tradeoff effect of ecosystems was more significant at small scales. In addition, the contribution of drivers to ESVs was quantified using a geographic detector model, and it was found that the combined effects of drivers were much higher than their individual effects. Therefore, the relationship between driving factors and ESVs should be fully considered in the construction of ecological civilization in the future. In particular, when formulating development policies, relevant departments need to find a balance between development and protection to achieve coordinated development of ecosystem services at different levels, and to ensure ecosystem stability while steadily increasing ESVs.

## Figures and Tables

**Figure 1 ijerph-19-08595-f001:**
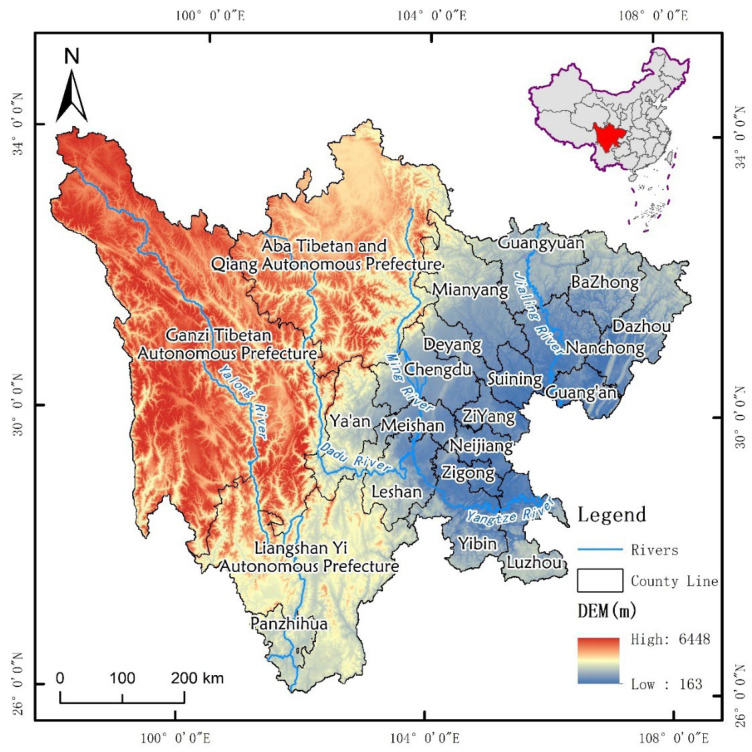
Location of Sichuan Province.

**Figure 2 ijerph-19-08595-f002:**
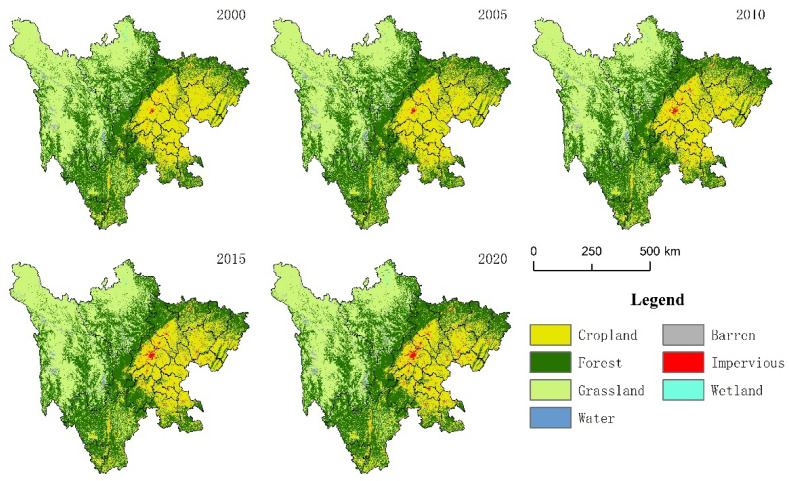
Spatial distribution of land use from 2000 to 2020.

**Figure 3 ijerph-19-08595-f003:**
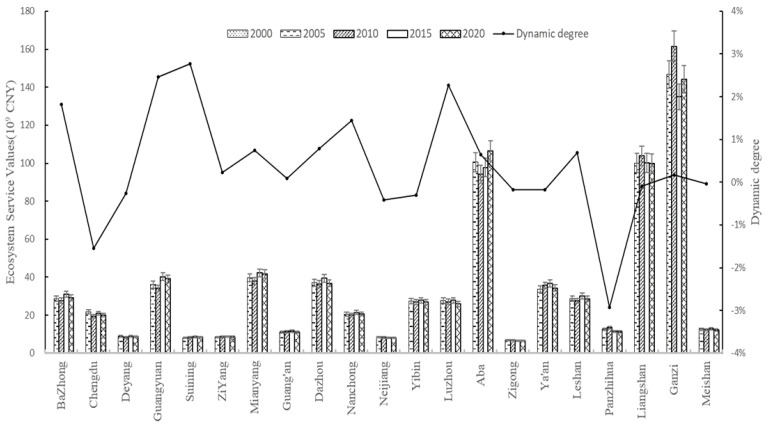
The value and dynamics of ecological waiters in various cities (autonomous regions) in Sichuan Province from 2000 to 2020.

**Figure 4 ijerph-19-08595-f004:**
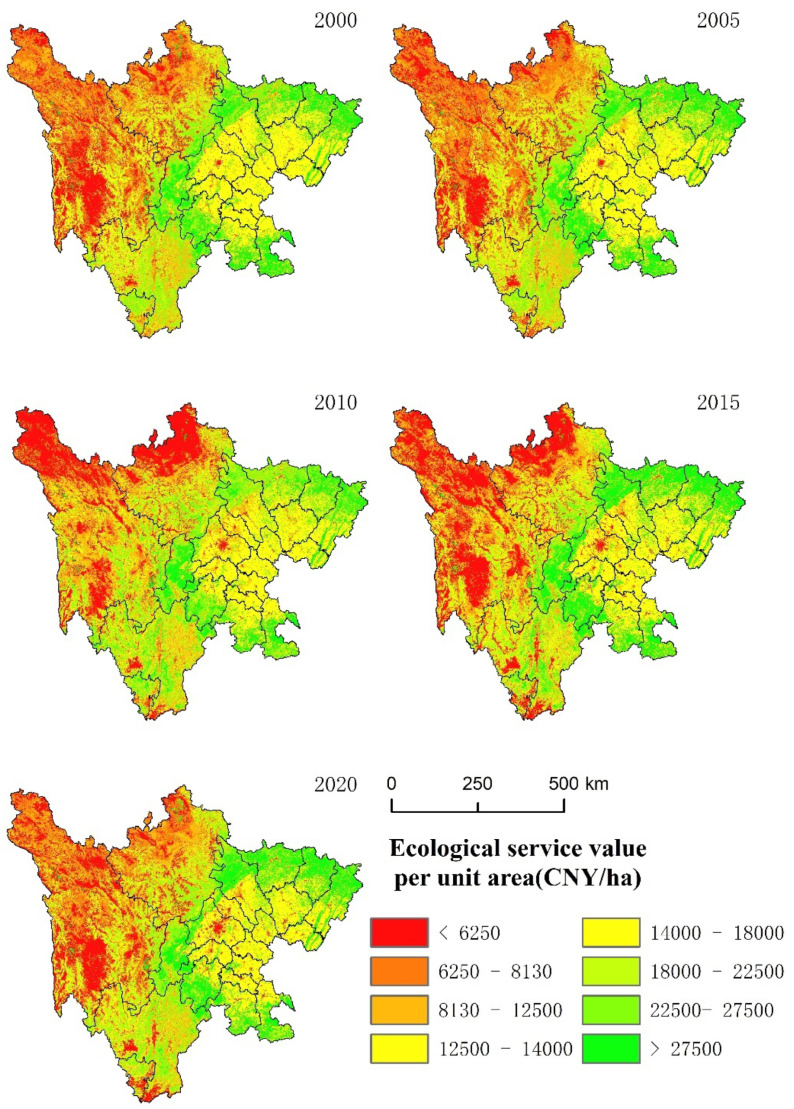
Spatial distribution of ESVs in Sichuan Province from 2000 to 2020.

**Figure 5 ijerph-19-08595-f005:**
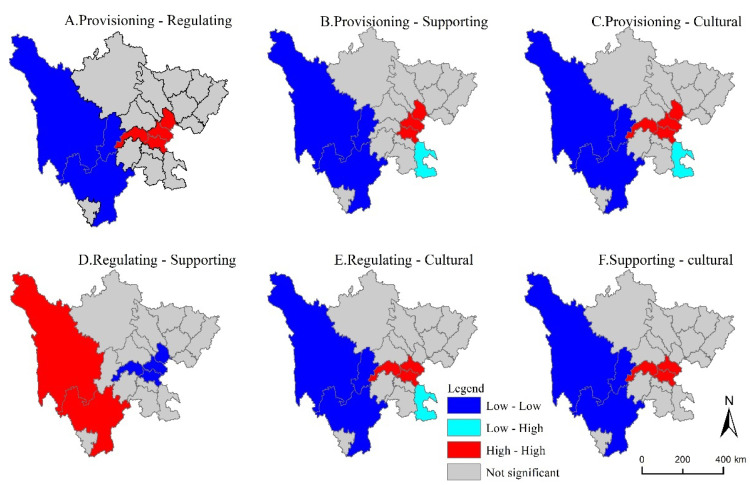
LISA cluster map of four ecosystem services in Sichuan Province at the city-scale.

**Figure 6 ijerph-19-08595-f006:**
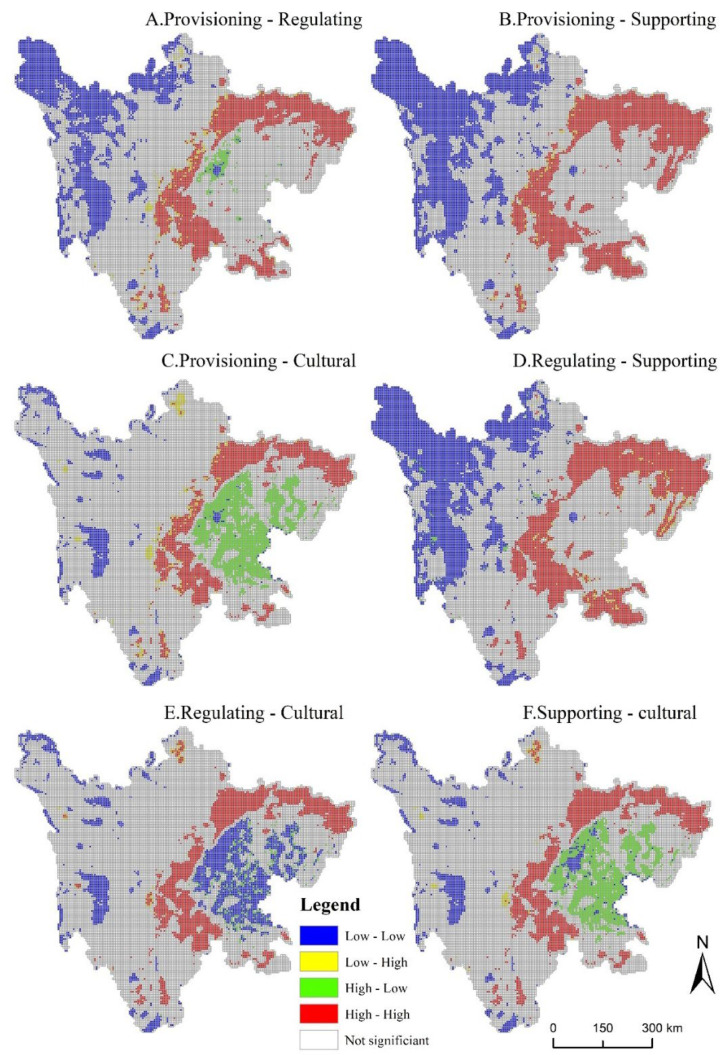
LISA cluster map of 4vecosystem services in Sichuan Province at grid-scale.

**Figure 7 ijerph-19-08595-f007:**
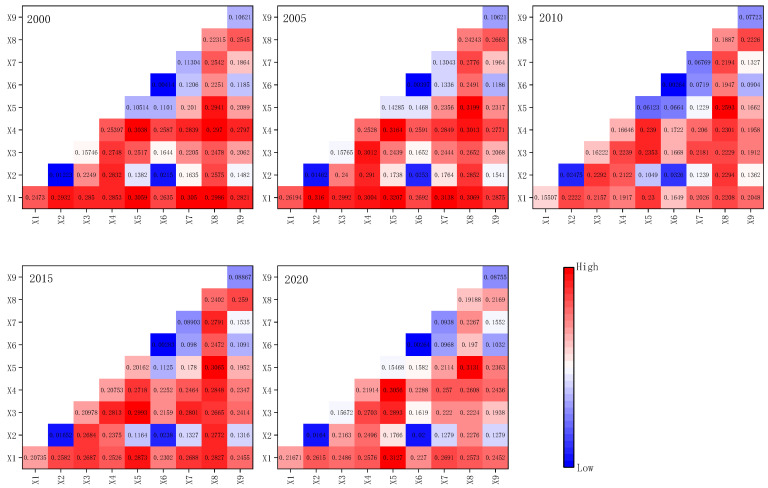
Interaction-driven results of ESVs in Sichuan Province.

**Table 1 ijerph-19-08595-t001:** Ecological service value per unit area.

Ecosystem Services	Farmland	Forestland	Grassland	Water	Bare Land	Construction Land	Wetland
**Provisioning services**							
Food production	1.35	0.205	0.1	0.1	0.01	0.01	0.51
Raw material production	0.135	0.475	0.14	0.01	0	0	0.5
**Regulating services**							
Gas regulation	0.675	1.555	0.51	0	0	0	1.9
Climate regulation	1.2015	4.65	1.34	0.46	0	0	3.6
Hydrological adjusting	0.81	3.345	0.98	20.38	0.03	−7.51	24.23
Environmental purification	2.214	1.385	0.44	18.18	0.01	−2.46	3.6
**Supporting services**							
Soil conservation	1.971	1.89	0.62	0.01	0.02	0.02	2.31
Biodiversity	0.9585	1.725	0.56	2.49	0.34	0.34	7.87
**Cultural services**							
Aesthetic landscape	0.0135	0.755	0.25	4.34	0.01	0.01	4.73

**Table 2 ijerph-19-08595-t002:** Land use area in Sichuan Province from 2000 to 2020.

Types	Areas/(km^2^)					2000–2020
	2000	2005	2010	2015	2020	
Cropland	120,296	118,517	117,373	116,143	111,919	−1.73%
Forest	190,412	192,560	194,622	195,452	200,371	2.06%
Grassland	163,397	162,129	160,408	159,285	157,634	−1.19%
Water	4033	4674	4957	4875	4234	0.04%
Barren	3280	3424	3181	3974	4677	0.29%
Impervious	1907	2313	2993	3833	4453	0.53%
Wetland	418	126	209	181	455	0.01%

**Table 3 ijerph-19-08595-t003:** Changes in the value of ecological services in Sichuan Province from 2000 to 2020.

Ecosystem Services	ESV (10^9^ CNY)					The Average Annual Increasing Rate	K_esv_
	2000	2005	2010	2015	2020	2000–2020	2000–2020
Food production	30.64	30.34	30.17	29.94	29.27	−0.89%	−0.76%
Raw material production	18.21	18.28	18.36	18.37	18.61	0.43%	0.36%
Gas regulation	64.73	64.87	65.11	65.08	65.71	0.30%	0.25%
Climate regulation	175.69	176.44	177.33	177.44	179.72	0.46%	0.38%
Hydrological adjusting	136.41	137.45	138.43	137.31	137.36	0.14%	0.12%
Environmental purification	94.26	95.39	95.86	95.05	92.88	−0.29%	−0.24%
Soil conservation	98.12	97.99	98.1	97.88	97.97	−0.03%	−0.03%
Biodiversity	77.21	77.32	77.74	77.7	78.34	0.29%	0.24%
Aesthetic landscape	28.86	29.24	29.62	29.6	29.85	0.68%	0.56%
Total value	724.13	727.32	730.72	728.38	729.71	0.15%	0.13%

**Table 4 ijerph-19-08595-t004:** Ecological service value and proportion of land use in Sichuan Province from 2000 to 2020.

Types	ESV/(10^9^ CNY)					Proportion/%				
	2000	2005	2010	2015	2020	2000	2005	2010	2015	2020
Cropland	157.42	155.09	153.6	151.98	146.45	21.59	21.14	20.79	20.57	19.75
Forest	427.05	431.86	436.49	438.35	449.39	58.56	58.87	59.08	59.34	60.59
Grassland	113.23	112.35	111.15	110.37	109.23	15.53	15.32	15.05	14.94	14.73
Water	25.92	30.05	31.88	31.34	27.21	3.55	4.1	4.32	4.24	3.67
Barren	0.19	0.2	0.19	0.23	0.28	0.03	0.03	0.03	0.03	0.04
Impervious	−2.57	−3.11	−4.03	−5.16	−5.99	0.35	0.42	0.55	0.7	0.81
Wetland	2.89	0.87	1.44	1.25	3.15	0.4	0.12	0.2	0.17	0.42
Total	729.26	733.54	738.77	738.7	741.69	100	100	100	100	100

**Table 5 ijerph-19-08595-t005:** Correlation of ecosystem services in Sichuan Province.

Correlation	Provisioning Services		Regulating Services				Supporting Services		Cultural Services
	Food Production	Raw Material Production	Gas Regulation	Climate Regulation	Hydrological Adjusting	Environmental Purification	Soil Conservation	Biodiversity	Aesthetic Landscape
Food production	1								
Raw material production	−0.268	1							
Gas regulation	−0.134	0.990	1						
Climate regulation	−0.290	0.999	0.986	1					
Hydrological adjusting	−0.179	0.480	0.461	0.491	1				
Environmental purification	0.283	0.034	0.063	0.046	0.790	1			
Soil conservation	0.527	0.678	0.772	0.660	0.271	0.228	1		
Biodiversity	−0.020	0.862	0.881	0.857	0.744	0.436	0.738	1	
Aesthetic landscape	−0.357	0.518	0.474	0.535	0.947	0.704	0.169	0.744	1

Note: Correlation is significant at the 0.01 level (2-tailed).

**Table 6 ijerph-19-08595-t006:** Global spatial autocorrelation of four ecosystem services in Sichuan Province.

**Ecosystem** **Services**	**Scale**	**Provisioning Services and** **Regulating Services**	**Provisioning Services and Supporting Services**	**Provisioning Services and** **Cultural Services**
Moran’s I	City-scale	0.435	0.482	0.362
	Grid-scale	0.377	0.669	0.01
**Ecosystem** **Services**	**Scale**	**Regulating Services and** **Supporting Services**	**Regulating Services and** **Cultural Services**	**Supporting Services and** **Cultural Services**
Moran’s I	City-scale	0.432	0.348	0.36
	Grid-scale	0.709	0.605	0.453

**Table 7 ijerph-19-08595-t007:** Detection results of driving factors for spatial differentiation of ESVs in Sichuan Province.

Driving Factors	2000	2005	2010	2015	2020
X1	0.247301	0.261944	0.155072	0.207354	0.216714
X2	0.012226	0.014617	0.024748	0.016525	0.016403
X3	0.157459	0.157646	0.162215	0.20978	0.156716
X4	0.253969	0.2528	0.166463	0.20753	0.219138
X5	0.105141	0.142849	0.061232	0.201623	0.154678
X6	0.004143	0.003967	0.002635	0.002828	0.002635
X7	0.113036	0.130435	0.06769	0.089033	0.093796
X8	0.223148	0.242428	0.188704	0.240205	0.191877
X9	0.106206	0.106207	0.077228	0.088671	0.087548

## Data Availability

The data presented in this study are available contained within the article.

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
