# Peer review of "Spatiotemporal Variation and Driving Forces Analysis of Eco-System Service Values: A Case Study of Sichuan Province, China"

_ijerph, 2022, doi:10.3390/ijerph19148595_

Round 1

Reviewer 1 Report

Review, Manuscript ID: ijerph-1720286

“Spatiotemporal variation and trade-offs/synergistic analysis of multiple ecosystem services in Sichuan Province”

This paper evaluates the temporal and spatial variation characteristics in the value of 9 selected ecosystem services from 2000 to 2020 in the case of Sichuan Province using GIS and Remote Sensing based analytical methods and provides insights into trade-offs and synergies between multi-scale ecosystems.

I recognise the strengths of this paper but also would like to highlight some weaknesses which I think need to be readdressed prior to publication.

Overall, there is a good idea behind this manuscript, but it needs to be more focused in terms of purpose, objectives, and outcomes. A stronger framework needs to be developed to provide a stronger structure for this manuscript.

In this context, I hope the following comments and suggestions help the authors to improve their manuscript and boost the quality of the paper.

  • Please can you ensure a clearer expression of methodology section in the manuscript, since there some unclear parts in it.
  • Please can you ensure that prior to resubmission you carry out a rigorous check of English grammar and language.

Introduction, aim and research questions

The paper starts with a thorough and interesting presentation of the ecosystem services and their value, concluding that the assessment of ESV is an important tool to inform sustainable development of the study area. However, I feel that you have failed to give why a correct understanding of the changes in ES vales is important especially in your study area and link it to your aim and objectives.

Please make sure that you have introduced all terms before using the abbreviation *please see attached pdf file.

Materials and Method

The Materials and Method are thoroughly presented. However, you should go through and try to explain each method that you have used in this study more clearly.

Also, I did not understand the order of methods you have used. I think you should have calculated ESs first, and then you should have looked at the changes in ESs. If so, please give the correct order for your study otherwise it is really hard to understand and follow what you have done.

Results

The results are thoroughly presented. But some parts needs to be gone through and rewritten to make it clearer.

Discussion

Discussion section should be shortened, in this form it is too long and so hard to concentrate on. 

Also I would expect to see some thoughts on the drivers of change in their amount and value of ESs as well as some insights into landscape and urban planning for enhancing ESs and their value in an urban environment.

I have also attached a pdf file containing some specific comments on your paper.

Author Response

Thank you for processing our submission and giving us the opportunity to revise. We have processed your comments and detailed all changes made to the manuscript.

Introduction, aim and research questions

The introduction of the article had been revised as a whole. The overall framework was: firstly, it introduced ecosystem services and ESVs, secondly, it introduced the evaluation method of ESVs, and then briefly analyzed the trade-off and synergy between ecosystem services. Then, it introduced the research progress of driving factors and ecosystem services, and finally, combined with the situation of the study area, explained why the relevant research was done, and provided a scientific basis for this article. The relevant terms were explained in the article (P43, P50), and the importance of studying the changes in the value of ecosystem services in Sichuan Province was proposed in the fifth paragraph.

Materials and Method

The order of this part of the method has been adjusted, first to calculate the ecosystem service value, then to calculate the dynamic degree of the ecosystem service value, and finally to analyze the ecosystem service value. In the data analysis, a geographic detector module was added to analyze the drivers of ecosystem service value. In Data sources and processing, data sources for driving factors were added (P170).

 Result

This part mainly added two parts: 1. Land use change; 2. Driving force analysis. Re-edit some incoherent parts according to the attachments.

 Discussion

We rewrite 4.1 of this part, not only discuss from the method, but also put forward corresponding opinions. At the same time, the results of the driving force analysis were discussed to make the overall structure of the article more complete.

Modifications for comments made in the annex(The front is the original manuscript, the latter is the revised version):

  1. P12:in the city (autonomous prefecture) scales and grid scales. --- P17:in the city (autonomous prefecture) and grid scales.
  2. P38:]; Ecosystem --- P45: ]; ecosystem
  3. P43:The diversity of ecosystem services and the uneven distribution of spaces lead to tradeoffs between services and mutually reinforcing synergies. ---    P70:There are various degrees of trade-offs and synergies among ecosystem services due to their complex and dynamic interactions
  4. P77: services[13]: One is ---   P55:ESVs [8]: one is
  5. P90:index[25, 26], ---  P68:index [22, 23],
  6. P92:RSEI and MNDWI to.,Deleted here, but later defined on first use --- The Modified Normalized Difference Water Index (MNDWI) and the Remote Sensing Ecological Index (RSEI).
  7. P95: EVs,This has been deleted.
  8. P120,P121,P123,Quantify,Identify,Use ---  P120,P121,P123,to quantify,to identify,to use.
  9. P125:It doesn't make sense to put it here, it has been deleted.
  10. P128:Located at ---   P128:Our study area is located at the
  11. P132:above 3km; The east ---    P133:. The eastern part
  12. P150,The determination of the equivalent factor of each land use type is described later, mainly based on the research results of previous papers, and there are references.
  13. P158:and 2020) were selected 158 based on the 30m resolution ---   P159:were obtained from the 30m resolution
  14. P164:The sudden line break issue has been resolved, but this has been removed.
  15. P172:Methods for Assessing Ecosystem Services ---  P179:Ecosystem Service Values
  16. P226:Use correlation analysis methods to determine whether there are synergies or trade-offs between those ecosystem services. ---   P234:Correlation analysis were conducted to determine whether there were synergies or trade-offs between these ecosystem services, according to Formula (8).
  17. P236:n multiple variables. Anselin et al.[37] proposed ---  P244:variables, Anselin et al. proposed a bivariate spatial autocorrelation based on the Moran index to reveal the correlation characteristics of the spatial distribution of different elements [67]
  18. P246:of various ecological services for ---  P282:According to formulas (1)-(6), the ESVs of seven land-use types in Sichuan Province from 2000 to 2020 were calculated.
  19. P444:Figure 6. Land use of Sichuan Province. ---  P273:Land use change was briefly described in the article.

Reviewer 2 Report

General comments:

The manuscript addresses an interesting and relevant topic; besides it add some interesting approaches in the methods of evaluating relationships among multiple ecosystem services.

Major concerns are:

Writing style throughout the manuscript is highly redundant which makes the reading and following the main ideas mostly difficult.

The discussion section must be improved. Despite the amount of results of the research, no mentions or analysis are on the reasons, possible implications, or driving forces behind the synergies and trade-offs found among the ecosystem services. Therefore, the discussion turns out very limited, sometimes even repeating the results. The authors have enough literature review that could enrich the discussion, thus improving the manuscript contribution.

Specific comments:

Lines 51-53: please review writing, duplicated sentence

Lines 92-. Please define RSEI and MNDWI since this the first time you use it

Lines 146: please explain at least some general aspects of the monetary value evaluation method

Lines: 159: please define CLCD

Lines: 168: please define RMB

Lines: 182: please define RS-based dynamic

Lines 211-223: please provide some references to support this method

Table 4: please provide significance level for each correlation coefficient, not enough to state that various had significance level of 0.01

Lines: 345: please review writing

Lines: 352: please review writing

Lines: 364-366: please review writing

Lines413-417: please review writing, also citation format

Lines 419-422: please review writing. What are the implications of these statements on the results of this study?

Lines: 424: please review writing, ¿what does “relatively comprehensive” means in this context?

Lines 426-428: please review writing. How these results relates to your findings in this study?

Lines: 431-442: Please review, information provided in this paragraph adds nothing to a discussion.

Author Response

Thank you for processing our submission and giving us the opportunity to revise. We have processed your comments and detailed all changes made to the manuscript. There are many changes in the introduction part of this revision. The Discussion section of the article has been enriched with the addition of an analysis of drivers of ecosystem service value and a brief analysis of land use change.

P51-53:This sentence is repeated, but the introduction was rewritten to delete it

P92 : After the article was revised, RSEI and MNDWI were first used in P164:The Modified Normalized Difference Water Index (MNDWI) and the Remote Sensing Ecological Index (RSEI)

P146:The monetary value evaluation method is partially introduced in the introduction, which is a bit redundant here and finally deleted

P159:The land use data of five periods (2000, 2005, 2010, 2015, and 2020) were obtained from the 30m resolution annual China land cover dataset (CLCD)

P168:In the revised article, P20 has been defined: (unit: CNY/kg, CNY = Chinese Yuan)

P182:Modified to considering the impact of temporal and spatial changes in ecological quality on ESVs, RSEI and MNDWI were chosen to correct the ESVs of each pixel at each moment, It is mainly used to select remote sensing indicators to revise the ESVs, so that it has temporal and spatial characteristics.

P211-223P:Li et al. corrected the equivalent factor pixel by pixel through RSEI and MNDWI, which can effectively display the temporal and spatial changes of the ecosystem service value of each pixel [27]. In order to better distinguish the ecological status between pixels, this paper introduces RSEI and MNDWI to construct e(Si,j,t_h ).

Finally, the RSEI obtained by normalizing  ranged from 0 to 1. The larger the RSEI value, the better the ecological condition [65, 66].

Corresponding references have been added as support

Table4:Add under Table5--Note: Correlation is significant at the 0.01 level (2-tailed).

P345:In the regulation services, except for the weak correlation among gas regulation, climate regulation, and environmental purification, other ecosystem services showed a significant positive correlation.

P352:To further understand the relationship between ecosystem services in Sichuan Province, we analyzed the bivariate spatial autocorrelation between six pairs of ecosystem services at the city (autonomous prefecture) and 5km grid scales based on GeoDa software.

P364-366:At the city (autonomous region) scale, there were trade-off effects between provisioning services and cultural services, regulating services and cultural services, provisioning services, and supporting services in Luzhou.

P413-417,P419-422,P424,P426-428,P431-442: It was mainly focused on the discussion of the first part. There was a problem with the overall structure of this part, so the discussion of this part was rewritten as a whole.

Reviewer 3 Report

The manuscript reports an exciting aspect of the Multiple Ecosystem Services in the Sichuan Province, a topic of high relevance. As a general comment, the manuscript is very restricted to the study area, from text to citations. Also, a thorough revision of the English language is required. Authors must consider some points to make it attractive to a general audience.

Title: I recommend making the title more general: Spatiotemporal Variation and Trade-offs/Synergistic Analysis of Multiple Ecosystem Services: a case study in China

In the Introduction, the order of paragraphs should be changed, following a more easy-to-follow thread. Terms and expressions like "trade-off in ecosystem services" and "Ecosystem Service Value" must be defined and exemplified at the beginning. Authors must include examples from other parts of the world – I could identify only three from outside China. 

I suggest reporting values in US dollars instead of yuan to make the results more accessible to international readers.

The sections material and Methods and Results are well prepared, although the language is difficult to follow in some parts and requires English revision. 

The discussion needs a lot of improvement. Please, do not start with the limitations of the methods. Bring this discussion to the end of the section. As it is now, the discussion focuses on the significance of the findings but brings only a few citations and arguments to the text. You should improve the discussion with other researchers and add international case studies to corroborate or highlight the innovation of your data.

Author Response

Reply to Reviewer 3

Thank you for processing our submission and giving us the opportunity to revise. We have made the following changes in response to your comments.

Title changed to: Spatiotemporal Variation and Driving Forces analysis of Ecosystem Service Values: A case study of Sichuan Province, China

The introduction of the article had been revised as a whole. The overall framework was: firstly, it introduced ecosystem services and ESVs, secondly, it introduced the evaluation method of ESVs, and then briefly analyzed the trade-off and synergy between ecosystem services. Then, it introduced the research progress of driving factors and ecosystem services, and finally, combined with the situation of the study area, explained why the relevant research was done, and provided a scientific basis for this article. The relevant terms were explained in the article (P43, P50), and the importance of studying the changes in the value of ecosystem services in Sichuan Province was proposed in the fifth paragraph.

P20 has defined CNY for articles. Taking into account the exchange rate will change, so not dollars but RMB

The order of this part of the method has been adjusted, first to calculate the ecosystem service value, then to calculate the dynamic degree of the ecosystem service value, and finally to analyze the ecosystem service value. In the data analysis, a geographic detector model was added to analyze the drivers of ecosystem service value. In Data sources and processing, data sources for driving factors were added (P170). And the language had been revised.

The discussion as a whole has been revised. The first part did not start from the method, but discussed the relationship between land use and ecosystem services, and put forward relevant suggestions. It also analyzed the driving factors of ESVs, and cited previous research results as a reference to analyze the rationality of the results. Summarizing the previous research results, I thought a possible innovation was to introduce the Modified Normalized Difference Water Index (MNDWI) and the Remote Sensing Ecological Index (RSEI) to correct the regional ecosystem service value.

Round 2

Reviewer 1 Report

Thanks for revising the paper. 

The paper is now much more improved and clear. 

I believe that it can be published now in its present form. 

Reviewer 3 Report

The manuscript has improved a lot and all points raised during review were considered on the revision.